# Fintechs and Institutions: A Systematic Literature Review and Future Research Agenda

Jorge Tello-Gamarra [1], Diogo Campos-Teixeira [2,*], André Andrade Longaray [2], João Reis [3] and Martin Hernani-Merino [4]

1    Department of Agroindustrial Engineering, Federal University of Rio Grande,
     Santo Antônio da Patrulha 95500-000, Brazil; jorgetellogamarra@gmail.com
2    Institute of Economic, Administrative and Accounting Sciences, Federal University of Rio Grande,
     Rio Grande 96203-000, Brazil; andrelongaray@gmail.com
3    Industrial Engineering and Management, Faculty of Engineering, Lusófona University and EIGeS,
     1749-024 Lisboa, Portugal; reis.joao@ua.pt
4    Department of Marketing and International Business, Universidad del Pacífico, Lima 15072, Peru;
     mn.hernanim@up.edu.pe
*    Correspondence: diogomc2_14@hotmail.com

**Abstract:** The growth of fintechs has exponentially modified the international financial system. These changes affect social mechanisms that regulate the performance of economic agents, generating the need to modify the current role played by institutions. Despite the clear relationship that exists between fintechs and institutions, studies exploring the details of this relationship are still scarce. The objective of this article is to propose a review and analysis of the current state of research on fintechs and institutions. To achieve this goal, a systematic literature review was conducted, with the selection and analysis of 123 documents published which were based on preestablished inclusion and exclusion criteria. The main results show the development of a framework that allows us to increase our understanding of fintechs and institutions; the identification of three propositions that serve as a guide to the institutional landscape in which fintechs operate; and finally the recognition of a research agenda.

**Keywords:** fintechs; institutions; transaction costs; regulation

## 1. Introduction

Studies published in the 20th century already integrated the perspectives of 'technology' and 'finance', projecting a potential revolution in the financial sector due to the introduction of new technologies and new forms of processing data in the financial system. Everything indicates that this projected financial revolution has arrived. Exponentially boosted by the digital economy of the past decades [1], this financial revolution gave birth to a new economic agent, the fintech [2].

Fintechs represent an object of modern research that has awoken the multidisciplinary interest of financial market researchers and managers [2]. In the literature, studies related to fintechs have been conducted with different focuses, such as: mobile internet [3]; cloud computing [4], bitcoin [5], legislation [6,7], cryptocurrency [8,9], digital platforms [10], blockchain technology [11], and institutions [12–14].

Regarding research on fintechs and institutions, the subject has been broached in the literature as follows: fintech integration in certain countries [8,14], the construction of a digital structure for the financial market [15–17], the strategic process of fintech incorporation by institutions [18,19], and the fintech supervision and security regulation process [20,21].

Although these studies have identified important points regarding some of the impacts the fintech phenomenon has on the institutional matrix, there is still a need for studies that permit a more detailed understanding of how institutions adapt to fintechs. This literature

gap serves as the motivation for the present article. Thus, this study aims to review and analyze the current state of research that correlates fintechs and institutions. To achieve this goal, the chosen method is a systematic literature review of articles published between 2001 and 2020.

This study is different from other studies on fintechs and institutions for three reasons: (a) it is the first study that directly links fintechs to aspects of institutional change; (b) it creates a framework that allows us to summarize the complex relationship between fintechs, institutional change, and the agents involved in the process; and (c) it presents future research trends regarding fintechs and institutions.

In the following section, we present the theoretical background about 'fintechs' and 'institutions'. Subsequently, the method followed in this study is described. Next, we present the results of this study. Then, a discussion of the results is presented, which includes a research agenda. The article ends with some final considerations.

## 2. Theoretical Background

### 2.1. Fintechs: Emergence and Conceptual Evolution

Through the advancement of information technology, various market niches have appeared with the intent of fulfilling their clients' various needs. In the financial market, an economic agent known as a fintech was formed with the purpose of fulfilling user needs, such as a reduction in transaction costs and personalized accessibility to services [22]. In this sense, the study of the fintech phenomenon has become increasingly popular, growing exponentially as fintechs have filled gaps pertaining to financial intermediation and technology [23].

Fundamentally, fintechs allow for new financial resources to be offered to customers, such as shared financing [24], transfers via application [25], and electronic contract signatures [26]. Researchers, both theoretical and applied, who study financial institutions have tried to adapt their operations and analyses to these technological innovations [27]. As such, fintechs represent financial innovations for users of different economic levels (consumers or institutions) [10]. The relationship between fintechs and users has evolved along with advances and technological innovations in the current digital environment [28]. However, due to the ample spectrum of activities that fintechs can encompass, there is still a lack of consensus in the definition of this economic agent [2,22]. The term fintech is derived from the neologistic sum of the words 'financial' and 'technology'. This term generically describes the connection between modern technology and commercial activities, that is, internet technologies in the financial services sector [29]. Due to the amplitude of this expression, the term fintech has been given different definitions (see Table 1).

**Table 1.** Definitions for the term fintech.

| | Source | Definition | Perspective |
|---|---|---|---|
| 1 | [9] | " ... are the primary actor to lead, manage, and respond to the formation of markets". | Agents of the financial system |
| 2 | [30] | " ... agents that interact with each other to provide a wide array of financial products and services to end customers". | Agents of the financial system |
| 3 | [23] | "Fintech is the use of technology to provide new and improved financial services". | Agents of the financial system |
| 4 | [31] | "innovative companies active in the financial industry making use of the availability of communication, the ubiquity of the internet, and the automated processing of information". | Firms |
| 5 | [32] | "FinTech is a technology that uses IT in the financial world. FinTech therefore refers to new technological solutions that will even initiate a revolutionary transformation in the world of finance". | Products and services |
| 6 | [33] | "Fintech are companies that are a new, special category of para-banks". | Firms |

**Table 1.** *Cont.*

| | Source | Definition | Perspective |
|---|---|---|---|
| 7 | [34] | " … financial industry that applies technology to evolve financial activities". | Agents of the financial system |
| 8 | [35] | "Fintech" denotes companies or representatives of companies that combine financial services with modern, innovative technologies". | Firms |
| 9 | [36] | "Recent advances in information and communications technology (ICT) have led to the rapid development and expansion of new and innovative financial services, often termed Fintech". | Products and services |
| 10 | [37] | "Internet finance, which is often referred to as 'digital finance' and 'fintech' outside China, was coined by Xie e Zou (2012). | Products and services |
| 11 | [38] | "*Fintech* is na economic industry composed of companies that use technology to make financial services more efficient". | Firms |
| 12 | [39] | "Technologically enabled financial innovation. It is giving rise to new business models, applications, processes and products. These could have a material effect on financial markets and institutions and the provision of financial services". | Products and services |
| 13 | [40] | "Fintech is a new sector in the finance industry that incorporates the whole plethora of technology that is used in finance to facilitate trades, corporate business or interaction and services provided to the retail consumer". | Agents of the financial system |
| 14 | [41] | "Driven by technological advances, new services models have developed in the financial industry which offer additional opportunities to customers. Under the common denominator 'fintech', these new business aim to challenge existing financial institutions by using technology to deliver value to the customer in na alternative way". | Firms |
| 15 | [42] | "Fintech is conceptually defined as a new type of financial service based on IT companies' broad types of users, which is combined with IT technology and other financial services like remittance, payment, asset management and so on. Fintech includes all the technical processes from upgrading financial software to programming a new type of financial software which can affect a whole process of finance service. Therefore fintech can improve the performance of financial services and spread the finance service combined with mobile environment". | Products and services |
| 16 | [43] | "Financial technology" or "FinTech" refers to the use of technology to deliver financial solutions. The term's origin can be traced to the early 1990s and referred to the "Financial Services Technology Consortium", a project initiated by Citigroup in order to facilitate technological cooperation efforts. | Products and services |
| 17 | [44] | "FinTech refers to innovative financial services or products delivered via technology". | Products and services |
| 18 | [45] | "Fintech refers to the application of technology within the financial industry. The sector covers a wide range of activities from payments to financial data and analysis, financial software, digitized processes and payment platforms". | Products and services |
| 19 | [46] | "Technology applied to financial services has a significant impacto n our daily lives, frim facilitating payments for goods and services to providing the infrastructure essential to the operation of the world's financial institutions". | Products and services |
| 20 | [47] | "Beside indirect financing via commercial banks and direct financing through security markets, a third way to conduct financial activities will emerge, which we call 'internet finance'". | Agents of the financial system |

Source: the authors.

When analyzing the different definitions proposed for the term fintech (see Table 1), we noted that this concept has evolved over time. In this sense, the term fintech was defined through three perspectives: (a) products and services, (b) firms, and (c) agents of the financial system.

From the perspective of (a) products and services, fintechs are seen as portfolio management tools, covering activities from payments to financial analyses, operated through software programs and digitalized payment platforms [45]. These programs and platforms are responsible for facilitating payments and providing financial infrastructure [46]. In the market, this infrastructure take the form of services that integrate technology with the financial sector, which can initiate a revolutionary transformation in the world of finance [32,42,43].

Furthermore, to understand fintechs as (b) firms is to define them as organizations that use technology to conduct financial operations with the intent of obtaining a profit [35]. These firms combine financial services with modern technologies to make financial services more efficient [38]. In this way, Szpringer [33] (p. 11) understands fintechs as a "new and special category for banks". These firms take advantage of the availability of communication, the omnipresence of the internet, and the automated processing of information to create innovation in the financial sector [31].

Finally, other authors have used more encompassing definitions of fintechs, seeing them as (c) the economic agents responsible for an institutional revolution in the financial system. Recently, Breidbach and Tana [9] highlighted that fintechs are 'key actors' in leading, managing, and responding to market formation. In this sense, Muthukannan [30] (p. 1) characterizes them as "agents that interact with each other to provide a wide array of financial products and services to end customers". Therefore, a fintech is understood as the use of technology to provide new and improved financial services [23]. These new services integrate different resources for financial operations, such as transactions without intermediaries, mobile payments, microfinances, and crowdfunding [48–50]. After conducting this search within the literature, it was possible to roll the various definitions of fintech into one.

In the present study, we define fintechs as: "Economic agents responsible for creating innovative products and services in the financial system through the integration of new technologies to minimize financial costs and transaction costs in the financial system". These actors in the financial market represent a powerful tool for a sustainable economic and institutional development [51,52]. In this context, fintechs have been responsible for creating a new transactional environment, which has reduced service delivery costs by up to 90%, according to Ventura et al. [53]. This study covers the inter-relation of institutions and changes brought on by the fintech phenomenon.

### 2.2. Institutions

Institutions exist to manage the behavior of different actors in the environment [54]. Studies that have evaluated different institutional perspectives have taken various approaches, including a sociological approach [55], a neoclassical approach [56], and that of a new institutional economy [57,58]. Since fintechs are agents that emerge within the financial system, we believe that North's perspective exhibits the characteristics that can best analyze institutions in light of fintechs.

North [57] explains that institutions are crucial in order to both restrict and encourage human activities and economic development. He stated that the market behaves like a 'game' and defined the institutions as 'the rules of the game'. Furthermore, he described organizations as 'players' and argued that adaptations to institutions and new market trends represent alterations to the rules of this game.

North [59] differs from neoclassical institutionalists in that he develops an approach that integrates social, political, and economic aspects. This approach considers the fact that institutions are not always efficient due to the complexity of the environment in which they are inserted. This imperfect environment in the market, with asymmetrical

information, causes institutions to operate based on imperfect beliefs and understandings. Since their understandings are imperfect, institutions are constantly undergoing a process of institutional change with the intent of maximizing their performance [57].

With the development of technology, as a part of the institutional change process, the financial market is shown to be an environment of constant adaptive transformations. Thus, the research regarding the role of institutions pertaining to changes in the financial system is a constantly evolving topic. DiMaggio and Powell [55] highlight that the normative processes that permeate institutional life specifically structure the interactions among different types of organizations, such as regulatory agencies, consumers (or resources and products), and key suppliers. In this manner, it is possible to specifically understand that currency, financial operations, transactional platforms, and monetary indexes are a part of the institutions that structure the financial system. As such, fintechs appear as a vector for adaptation to different technological and informational resources that emerge in the market [59], providing access to informational and financial resources through their products and services.

North [57] describes the process of institutional change in the trajectory leading to a change in the market's relative prices. Thus, an incentive structure appears on behalf of the market and the responsible institutions with the aim of incorporating new technologies and processes. This incentive structure is responsible for generating tacit knowledge for institutions, leading to a phenomenon known as institutional learning. This phenomenon automatically affects the habits and customs of this market's operators, forming a new environment with informal rules. When these informal rules are escalated and perfected, this forces the regulatory institutions to formalize them. The changes in institutional rules allow entrepreneurs to explore more lucrative routes to maximize their profits and, consequently, increase economic performance in general (see Figure 1).

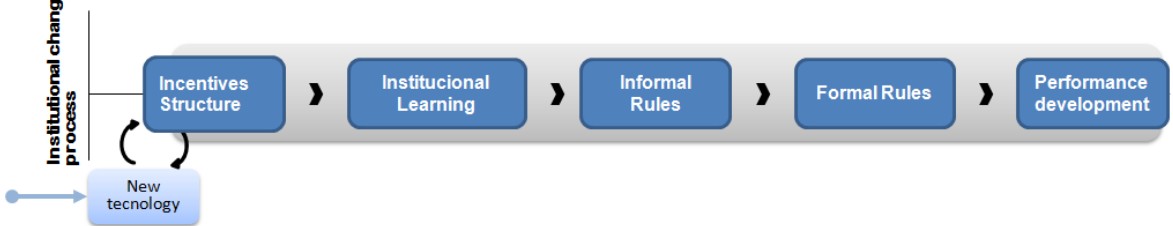

**Figure 1.** Institutional change process. Source: the authors.

Upon evaluating the role played by fintechs in this process, Li et al. [5] point out that fintechs generate market adaptations and can be responsible for a revolution in the financial system. In this sense, the interaction between the different 'players' and the new technologies creates space for researchers to try and understand the 'rules of the game' that have been adapted by fintechs. Furthermore, it is important to understand the actors involved in this process, since there can be circumstances in which certain organizations operate as institutions due to the range of the interactions between the organizations themselves [58]. Thus, since they eliminate intermediaries and improve financial operations, fintechs automate the activities of certain organizations [48]. As a result of the revolution provoked by fintechs, there is a gap in the understanding of how the financial system will react in order to manage the modifications occurring in these emerging conditions [59]. This makes it necessary to conduct a systematic literature review.

## 3. Method

To systematically evaluate the body of literature pertaining to this subject of research, we chose to undertake a systematic literature review. In this sense, in this systematic review, we analyze research that explores the interrelation of fintechs and institutions. The research execution strategy was based on the method proposed by Transfield et al. [60],

which includes the following steps: (i) planning, (ii) search, (iii) triage, (iv) extraction, and (v) summary of the results.

*Planning*. In this stage, we defined the research question, which was, "How does the literature report on the relationship between fintechs and institutions?". The answer to this question guided the next steps of the study. To ensure the consistency of these steps, we defined a set of inclusion and exclusion criteria. Therefore, the papers were selected according to these criteria.

*Search*. To answer the research question, we used the inclusion criteria described in Table 2. Through the main search strategy, we identified articles, editorials, and reviews by placing the "AND" operator between the term "fintech" and the following terms: "Institution", "Transaction cost", and "Regulation". The chosen databases were Scopus, Web of Science, and Science Direct. These were selected due to their credibility and their document selection criteria.

**Table 2.** Inclusion and exclusion criteria for the documents used in the systematic review.

| Criterion | Criteria Dimensions |
|---|---|
| Inclusion | (1) The document is located in the SCOPUS, Web of Science, or Science Direct database. <br> (2) The document contains the terms "Fintech" and "Institution", which are simultaneously cited in the title, abstract, or the keywords. <br> (3) The document contains the terms "Fintech" and "Transaction Cost", which are simultaneously cited in the title, abstract, or the keywords. <br> (4) The documents contain the terms "Fintech" and "Regulation", which are simultaneously cited in the title, abstract, or the keywords. <br> (5) The document was published between 1 January 2001 and 31 December 2020. |
| Exclusion | (1) The document is not completely written in English. <br> (2) The document does not count as an article, editorial piece, or review. <br> (3) The document is more than 20 years old. <br> (4) The document is not included in the areas of "Business, Management and Accounting", "Economics, Econometrics and Finance", "Social sciences", or "Decision sciences". <br> (5) The document does not fulfill the relevance criterion, which includes availability, methodological limitations, relevance of findings, and coherence. |

Source: the authors.

*Triage*. To guarantee research consistency, the studies that were found were selected according to a sequence of criteria. The exclusion or inclusion of the collected documents followed the criteria presented in Table 2. The triage considered factors such as the date, field of study, and the use of the English language. To delimit the selected research areas, we took into account criteria of proximity to managerial, economic, and social issues related to institutions, removing papers focused on the areas of the exact sciences, engineering, computer sciences, etc. In addition, the relevance of the paper was taken into consideration. Therefore, this method of analysis was attentive to understanding the contextual meanings of these studies and their practical objectives.

*Extraction*. The extraction process was conducted in two stages. First, a search was conducted to determine which concepts were more significant to the matters in question. Next, these concepts were grouped together with the intention of finding 'theoretical knots' between the different studies. Due to the data collected, and the qualitative nature of the categorization process, we built a theoretical framework to obtain amplitude in the results. The information collected from the documents was divided and computed as research data, as shown in Figure 2, using a total of 123 documents.

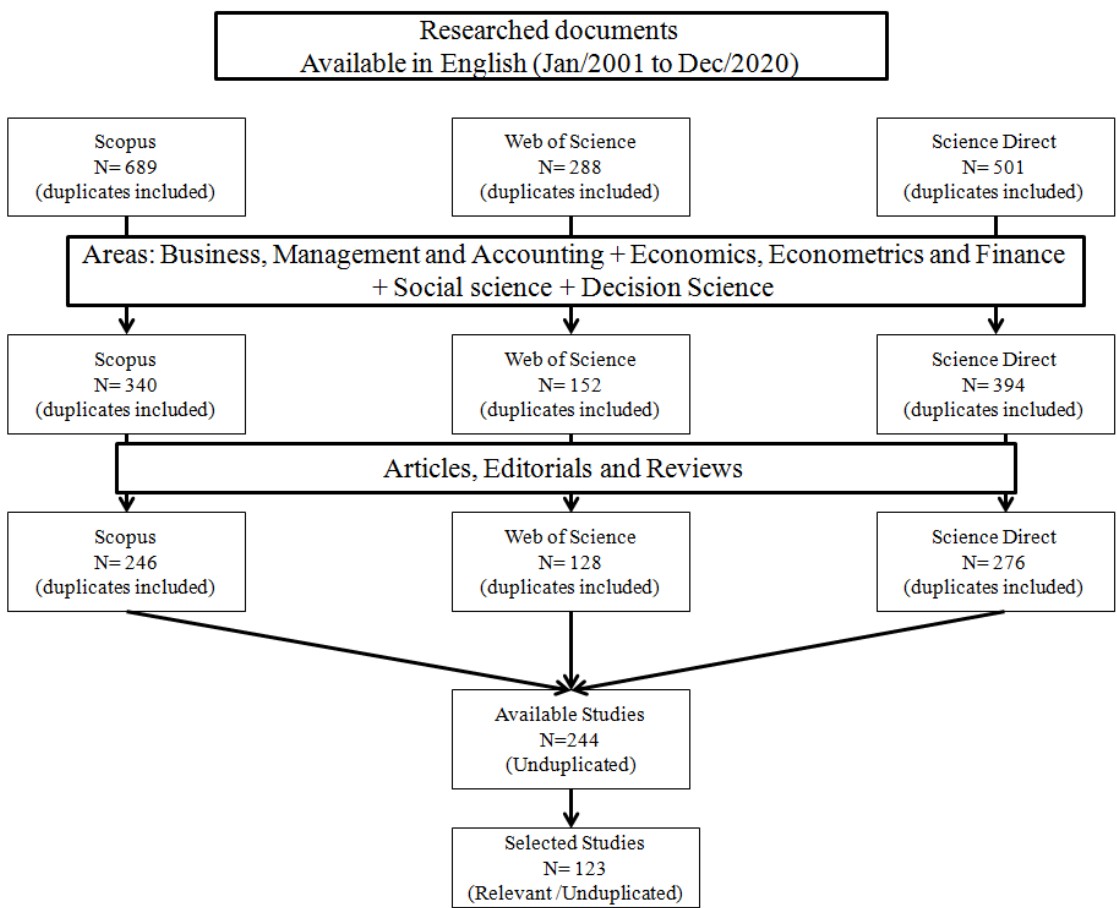

**Figure 2.** Selection criteria for the analyzed documents. Source: the authors.

*Summary*. We summarized the information that was found in order to highlight theoretical convergence points in the documents. It is important to mention that in the execution of this process it is necessary to interpret the analyzed documents subjectively. The data synthesis is the most value-added result of a literature review, as data analysis and synthesis can produce new knowledge [61]. Based on the described method, it was possible to group the conceptual results obtained in the review and describe a framework with an analytical approach to the studies that were developed. Through this process, we elaborated a conceptual consolidation of this theme, with theoretical nodes that explained the evolutionary changes in institutional theory brought about by the fintech phenomenon.

### 4. Results

#### 4.1. Stages of Institutional Change

After conducting a literature review on the multiple aspects related to fintechs and institutions, we discovered that fintechs are responsible for a process of evolutionary change in financial institutions. In the literature, it was possible to observe that the analyzed documents fit into four dimensions, which we propose as follows: dematerialization of access, operational architecture, transactional regulation, and transactional efficiency (Figure 3). These dimensions appear in the financial system as follows:

The *dematerialization of access* dimension includes fintechs that increase the capability of users to execute financial transactions without the presence of intermediaries, such as currency or physical banks and brokerage agencies. Furthermore, this dimension emphasizes the need for institutions to democratize financial services and promote financial inclusion [62]. The benefits brought about by fintechs affect both the clients and the institutions that are involved thanks to the increase in financial service accessibility and the reduction in transaction cost and time [63]. This dematerialization process requires

economic, political and social measures to be implemented [64]. Therefore, the modification of the use of financial services is a factor that affects the modification of the operational architecture of the financial market [65].

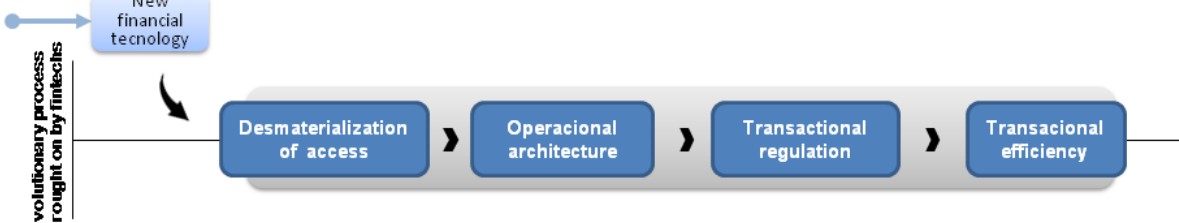

**Figure 3.** Stages of the evolutionary process brought on by fintechs. Source: the authors.

The *operational architecture* dimension, which is related to how financial operations happen, evolves in such a manner as to make transactions easier. In this sense, the development of financial intermediation technologies affords the agents of the financial system an increase in their operational capacity [66]. These technologies build a new structural paradigm in the financial market, which de-bureaucratizes processes and integrates resources for the consumer [67]. In this way, legal measures to eliminate intermediaries are being implemented with cloud technologies, blockchain and big data, artificial intelligence, biometrics, and application program interfaces which modify the financial system architecture [68]. These technologies, when incorporated by regulatory institutions, cause traditional regulations to no longer make sense; therefore, transactional regulation becomes increasingly technological.

In the *transactional regulation* dimension, the regulatory institutions act upon the financial system with measures that protect the operations from the risks related to operational modifications brought about by fintechs [26,48]. These modifications cause regulatory institutions to incorporate technological resources to control financial operations. This in turn enables regulatory flexibilization on the part of the institutions [69]. This factor results in a more managerial and less legislative position for the responsible institutions [7].

Finally, the *transactional efficiency* dimension highlights fintechs as a necessary tool for better functioning financial operations. The financial operations conducted by fintechs are more precise and flexible in their execution. The balance between this better functioning and a more open financial regulation permits the appearance of new concepts in the economy and new business models [19,24,25]. These concepts introduced by fintechs are responsible for generating new business models focused on payment models, wealth management, funding, and other financial services [1].

In the group of analyzed documents, it was possible to note that the body of literature pertaining to institutions and fintechs is still under construction. In the literature published up to 2014, there were no documents on the subject of fintechs and institutions because this phenomenon is recent and its validation began in the past 10 years. In this study, we listed the documents found in the systematic review in Appendix A. These documents exhibited different aspects of the four dimensions, which we have summarized in Table 3.

### 4.2. Dematerialization of Access

The first dimension identified in the literature is the dematerialization of access to financial resources through fintechs. This dimension exists due to the interaction of a structure of incentives with new financial technology. In this dimension, the following variables appear: *digital literacy* [70,71], the *democratization of access* to financial services and products [72,73], and the *financial inclusion* offered by fintechs [74–76].

Junger and Mietzner point out that fintechs are not only a cheaper way to interact with clients. On the contrary, they highlight that fintechs include factors pertaining to the *digital literacy* variable. This variable is related with exposure to technology, age, security, financial experience, and an emphasis on transparency [65]. In this sense, Gabor and Brooks [73]

stated that consumers are submerged in a digital revolution and, consequently, will adapt to the new operations brought on by the fintech phenomenon. In addition, they highlight that financial organizations need to pay attention to the digital literacy of users so as not to become obsolete, as consumers are increasingly looking for new opportunities in the digital environment [70]. To protect the consumer in this environment, Langenbucher highlights the application of anti-discrimination laws, which guarantee the integrity of data and operations to democratize access to consumers [71].

**Table 3.** A summary of the dimensions and the systematic literature review.

| Dimensions | Definitions | Documents |
|---|---|---|
| Dematerialization of access | Democratic access to financial services, with digital literacy and inclusion | (8) (19) (20) (23) (24) (26) (28)–(30) (46) (59) (65) (66) (83) (96) (99) (101) (111) (115) (117) |
| Operational architecture | Operational routines in the financial system | (5) (6) (7) (9) (11)–(15) (17) (20) (22) (26) (27) (33) (35) (39) (44) (47)–(49) (51) (52) (54) (56)–(58) (60) (62) (67)–(70) (75) (82) (85) (91) (94) (95) (98) (100) (102) (105) (110) (113) (116) (118)–(120) (122) |
| Transactional regulation | Laws and rules that are formalized for the liquidation of financial transactions | (2) (3) (7) (11) (13) (15) (16) (25) (26) (32) (36)–(38) (41)–(43) (46) (47) (54) (55) (64) (72)–(78) (81) (86)–(89) (99) (107)–(109) (112) |
| Transactional efficiency | Improving dematerialized financial operations and using risk management | (1)–(4) (6) (10) (15) (18) (20) (21) (26) (27) (29) (31)–(34) (36) (37) (40) (42) (44) (50) (53) (58) (61) (63) (66) (69) (75) (79)–(81) (84) (87) (90) (92) (93) (97) (103)–(106) (109) (114) (121) (123) |

Note: key in Appendix A. Source: the authors.

Haddad and Bratianu highlight the *democratization of access* to various financial resources in the digital era [64]. They explain that the dematerialization of bonds, money, and contracts democratizes consumer access to these financial products. Consequently, these products provide benefits regarding accessibility and cost reductions, making access to financial products more democratic [72]. In this sense, Cuttino [77] highlights the solutions provided by fintechs that are focused on low-income people, for example, promoting early salary transfer services, through mobile platforms, algorithmic technology, and GPS tracking. The emergence of new institutions resulting from these new technological advances undoubtedly affects existing institutions [78]. However, Omede understands that these digital resources were necessary as an alternative to various financial crises and currently have taken a leading role in the delivery of financial services to users from different social strata [72]. He understands that this market adaptation offers the integration of low-capital consumers into a huge portfolio of financial products, which is beneficial to the market as a whole.

Considering this, another variable is *financial inclusion*. Ozili [62] (p. 333) states that "Financial inclusion is a strategy to eliminate or reduce poverty". He points out that digital finances have played a crucial role in different areas for the inclusion of a larger group of consumers in financial operations. Digitized financial service providers have shown that they are meeting the demands that banks do not, in order to assist in the financial inclusion process [75]. Kotarba [76] confirms other factors that impact financial inclusion, such as transparency, wide competition, the digitalization of operations, lower cost complexities, risk sharing, and risk-based pricing. The development of these factors provides the opportunity to provide credits and loans within the platforms to a group of consumers that was not previously served [74].

Among the studies that analyze the factors that influence the consumer to make use of fintechs, the literature shows that the indicators of digital literacy, democratization of access, and financial inclusion are variables responsible for influencing consumer satisfaction and loyalty. This dimension of dematerialization of access is responsible for encouraging the learning of these new processes in the institutions involved and, consequently, modifying the operational architecture of the financial system.

### 4.3. Operational Architecture

In the studies that concern institutions and the payment ecosystem, the operational architecture dimension stands out. Within this dimension, the theoretical grouping highlights the following variables: adopting the *decentralized payments* ecosystem [63,65,67], *financial incentives* [21,79], the *elimination of intermediaries* [1,80,81], and *smart contracts* [48,82,83].

Shkarlet et al. [67] emphasize this change in client services due to the introduction of *decentralized platforms*. This variable helps to improve the quality of bank settlement and the provision of resources to the financial system's actors [84]. Peer-to-peer platforms can change the configuration of different banks and their ways of providing financial services [75]. Petrushenko et al. confirm that digital technologies mitigate market imperfections and provide innovative services that fulfill the needs for speed, low cost, security, and transparency [63]. Additionally, financial service providers can learn useful lessons from the experience of companies operating in a decentralized way, which have characteristics related to high performance and the ability to reinvent themselves, which Fenwick and Vermeulen characterize as 'innovation ecosystems' oriented towards finance [79].

In this sense, financial systems around the world have introduced the variable of *financial incentives* to adapt their financial operations. As such, Tarkhanova et al. pointed out various changes in the operational architecture of organizations, such as computers, ATM machines, mobile payment resources, and internet banking [21]. Additionally, the peer-to-peer market brought about by fintechs has been dominated by investors and banks, and this factor attracts capital to the sector. In this sense, institutions have promoted incentives related to consumer protection measures within internet layers [84,85]. The literature clarifies that the purpose of these changes is to promote the development of financial systems and provide alternative sources of cross-border funding, loans, and investments [86].

These technological resources encourage the emergence of the *elimination of financial intermediaries* variable. Thus, resources such as currency, contracts, bonds, and products have been reaching the final consumer in digital form, without intermediation, thanks to the presence of fintechs [80]. The disruptive effect of fintechs in providing these resources was to focus on a system based on trust in financial institutions, or to explore other environments. However, without intermediaries for the execution of financial transactions, trust becomes indispensable [87]. In this sense, the innovations provided by fintechs present challenges for financial authorities, as these technologies make it possible to eliminate financial intermediaries. Therefore, challenges regarding data management, integration, and privacy emerge, due to the fact that these data are decentralized [1].

Additionally, in our review of the literature we highlighted the variable of *smart contract* creation. Haddad and Bratianu [64] go into detail about the benefits of dematerializing contracts. Together with the digitization of money, the digitization of contracts brings about advances in the market in terms of transaction costs [88]. Furthermore, Brammertz and Mendelowitz demonstrate the importance of digitalizing financial contracts to standardize cash flow, despite also pointing out that this can represent risk [83]. Liu et al. summarize studies on risk, assessing the credit and market risk prospects in new operations based on digital contracts in the financial system [89]. In addition, the authors propose solutions such as diversification and transfer to control these risks.

To mitigate these risks, the literature suggests the need to implement new rules in the financial system. In this way, a new regulatory model emerges on the part of the institutions involved, a factor that encourages the emergence of the transactional regulation dimension.

### 4.4. Transactional Regulation

In the transactional regulation dimension, the agents of the financial market adapt the financial system's regulation to the fintechs. Thus, the literature pertaining to the transactional regulation dimension contains the following variables: *development of regulatory technologies* [17,80], *regulatory platforms* [13,90,91], and the *integration of regulatory authorities* regarding fintechs [17,24,29].

In the *development of regulatory technologies*, institutions start to use technology to safely handle digital transactions. Different authors have highlighted a subgroup of fintechs responsible for regulating financial operations, known as regtechs [17,80,92]. Arner et al. understand fintechs as resources that enable companies to comply with the current regulations in a more prudent way [17]. In addition, the literature indicates that regtechs are vehicles that help the financial system's regulatory authorities to apply regulations more efficiently and supervise financial organizations [43,80]. However, the fintech phenomenon still challenges some regulatory structures due to its rapid change [93]. Therefore, a regulatory approach that integrates technological resources represents unexplored territory, which is likely to contribute to the future development of governance in the financial market [94].

In relation to the *regulatory platforms* variable, various authors point out the importance of the appearance of integrated and decentralized business models for the evolution of the financial system [13,90,91]. These business models represent a challenge for regulatory authorities, in the sense of achieving a balance between the innovation brought about by fintechs and the prevention of risks to financial stability [93,95]. Micheler and Waley mention the regulatory uncertainty of these business models, although it is possible to anticipate the problems that will appear as the technology evolves [96]. Chen and Bellavitis highlight potential challenges that these platforms can represent, such as volatility, fraud vulnerability, and unstable regulation [13]. Some of these platforms guarantee their credibility through the use of tokens and cryptoassets that ensure the settlement of their transactions [97].

Consequently, this process is associated with the *integration of regulatory authorities* variable. Many authors [24,98] have explained the effectiveness of more flexible legislation. They point out that a regulatory system that uses technological instruments to guarantee transactional safety can represent the next step in the financial system's evolution. Additionally, a range of authors [99–101] have demonstrated that fintechs in countries with weaker regulatory institutional structures have greater room for growth. In this sense, several authors [29,102] have highlighted the authorities' position in the transactional model with the presence of fintechs. Gomber et al. point out that authorities will only maintain their control over transactional media through technology-based regulatory mechanisms [29]. Therefore, the self-regulation which integrates technology with the authorities demonstrates a tendency to control the risks generated by fintechs [103]. For the implementation of these technologies, researchers have suggested regulatory sandboxes to help authorities build a regulatory structure that promotes financial innovation and market security [17,24,94,104].

From this perspective, it is also possible to note the integration of technologies, platforms, and authorities in a technology-based regulatory model [26]. In this sense, this dimension proves to be important for the process of formalizing the rules of the financial system, in which technological resources are used to reduce risk. These rules are oriented towards the evolution of the financial system. This evolution involves a dimension aimed at increasing the financial system's efficiency.

*4.5. Transactional Efficiency*

The final dimension refers to the increase in transactional efficiency. This dimension explains how technological resources can be inserted in the financial system quickly and securely. The literature on this subject mentions variables pertaining to the *shared economy* [89,105], *data security* [7,11,16], and the use of technologies for *risk management* [70].

The *shared economy* variable represents operations associated with risk sharing in financial operations. Risk sharing is the method responsible for reducing uncertainty in operations [19,24]. Consequently, the reduction of uncertainty reduces the costs required to execute a financial operation [106]. Liu et al. emphasize that with information asymmetry, the shared economy makes it possible to conduct high-risk operations with less uncertainty [89]. With the reduction of uncertainty in these operations, the offer of financial products and services becomes less expensive. Staikouras highlights the regulation of platforms that promote the sharing economy in a cross-border way, with a commitment

to reaching parts of the market that have not yet been reached [107]. In this way, through these platforms, the frequency in terms of the acquisition of financing and investments in certain assets grows in a collective way [108]. This introduces the promise of many benefits, including efficiency growth and reduced costs for businesses, consumers, and intermediaries [109].

Another variable in this dimension is *operational security*. The literature points out how difficult it is for the traditional regulatory system to identify cybernetic crimes [7]. In this sense, this variable is associated with the introduction of cybersecurity variables to ensure the security of data and operations [11,16]. Huang et al. highlight several risks related to liquidity, security, and data privacy [110]. The researchers point out the applicability of cybersecurity features to ensure the security of operations. These features aim to protect both banks and public institutions from money laundering and fraud-related issues [111].

Here, we have the emergence of the *technological risk management* variable. With the growth and formalization of fintechs, the risk management model for the services offered by fintechs became mostly digital. Thus, fintechs provide services such as payments, transfers, loans, asset management, planning, and insurance—all thanks to technological management. In addition, fintechs are incorporating new technological concepts into the financial system, such as cloud technology, artificial intelligence, and big data management. In this way, these technological resources increase the proximity of their customers to financial services, since their services are made available remotely and with the minimization of risks [70]. Giudici highlights the role of regulators in the process of technological risk management [112]. Furthermore, Golubic highlights the challenge of implementing adaptable rules in the financial system, which can mitigate risk and accompany the speed of evolution of the delivery of these services to customers [75]. Due to the difficulty of finding this balance, some countries have invested massively in risk reduction alternatives through technology and have consequently developed an adaptive regulation approach [113,114].

In this way, it is possible to observe that fintechs represent agents that help to increase the transactional efficiency of the financial system. This dimension clarifies that a new product or service has been effectively implemented in the financial system and contributes to increasing the performance of a given economy. In Table 4, we present the four dimensions identified in this systematic review. Each dimension is presented alongside the variables that comprise it.

**Table 4.** Dimensions and variables of institutions and fintechs.

| Dimensions | Variables | References |
|---|---|---|
| Dematerialization of Access | Digital literacy<br>Democratization of access<br>Financial inclusion | [62–65] |
| Operational Architecture | Decentralized operations<br>Financial incentives<br>Elimination of intermediaries<br>Creation of smart contracts | [13,26,66,67] |
| Transactional Regulation | Development of regulatory technology<br>Regulatory platforms<br>Integration of regulatory authorities | [7,26,29,69,98] |
| Transactional Efficiency | Shared economy<br>Operation security<br>Technological risk management | [1,19,21] |

Source: the authors.

## 5. Analysis of Results

### 5.1. An Integrative Framework

In the analyses that were found pertaining to fintechs and institutions, we detected that the evolutionary process of institutions in the 21st century includes adapting to fintechs. This adaptation consists of restructuring the operations the institutions execute using technology. For this purpose, the literature presented the need for a structure of incentives for financial institutions to use new technologies. This structure of incentives was shown to include different structural measures linked to innovation in the transactional model, including electronic signatures and cryptography. These resources dematerialize consumer access to financial services and products.

The process of dematerializing access is linked to the interaction among different agents of the financial system. This interaction generates financial literacy and inclusion due to the use of technologies in the financial system. In this sense, the institutions go through a process of technological learning to democratize access to these financial and technological resources. This adaptation generates a change in the financial system's operational architecture.

This change in the operational architecture is associated with the incorporation, investment, and scalability of operations without intermediaries. Thus, new contract models are developed, and some intermediaries become obsolete. As such, the rules of the financial system are informally changed. This change involves the variables of the operational architecture dimension and awakens the need to regulate this new transaction model.

After the new rules are in place, the need arises to formalize the transactional regulation. Thus, these new rules modify the financial system's institutional structure. In this sense, the authorities centralize the regulation in digital platforms, which use new cryptography technologies to guarantee the integrity of financial operations. Regulatory fintechs (regtechs) and regulatory sandboxes are tools that make this possible. However, new measures are needed to improve the efficiency of this new operational model in financial institutions.

Finally, integration between authorities improves transaction efficiency. This efficiency is necessary so that the financial system's institutions can safely incorporate the resources provided by fintechs. Thus, risk management measures, an increase in operation frequency, and the centralization of information improve financial operations. Risk sharing reduces operation and transaction costs, as it reduces transactions' uncertainty. Reducing uncertainties improves economic performance when providing services to consumers. We call this entire process the evolutionary process of institutional change brought on by fintechs (see Figure 4), which can be understood through the following framework.

To efficiently execute the fintech incorporation process, institutions must interact with other actors, such as technology suppliers, other financial institutions, and users from different dimensions of the process. Within this study's perspective, fintechs are responsible for intermediating the access to financial technologies for the consumers of financial products in a decentralized manner. This intermediation reduces costs and stimulates more financial incentives. In addition to this framework, we also present a typical discussion with insights about the dimensions identified in this review, and then make some propositions.

### 5.2. Insights about the Dimensions

In the analysis of these studies, it was possible to observe that the body of literature that integrates the perspectives of institutions and fintechs is still under construction. In this sense, the literature addresses specific aspects of this relationship indirectly. In documents developed before 2014, this relationship was even more superficial and ignored relevant aspects between institutions and fintechs. Therefore, based on this review, it was possible to develop important insights regarding the dimensions highlighted in this study.

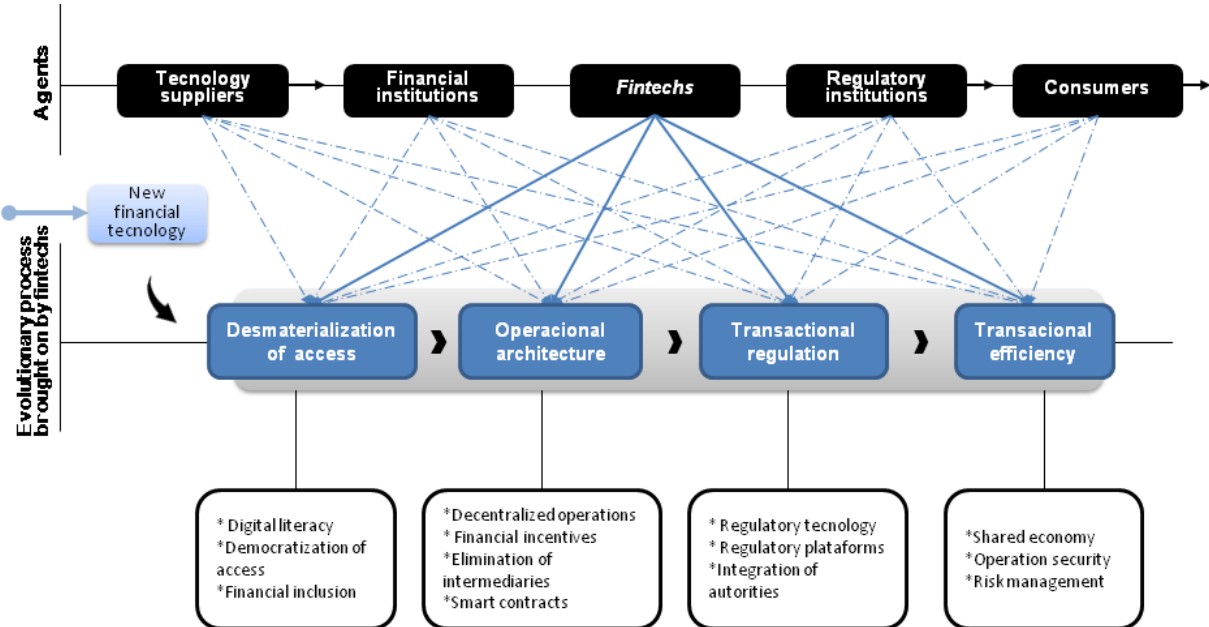

**Figure 4.** Framework regarding institutions and fintech market agents. Source: the authors.

### 5.2.1. Dematerialization of Access

Initially, in an institutional analysis, it is understood that a process of institutional change starts from an incentive structure. This incentive structure can represent funding, public policies, or social movements that interact with each other so that an innovation can enter the market. When it comes to fintechs, this incentive structure culminates in the dematerialization of access to financial products and services.

In this case, as Hodson notes, fintechs represent a vector for a new dematerialized policy of transactions [105]. The author calls fintechs the 'Uber of banks', because they provide an increase in the frequency of investments in specific assets due to their accessibility. This accessibility develops a forced user learning process, in which the user eliminates steps in the process of acquiring and using financial products and services through digital literacy [90]. In addition, institutions in the sector have promoted initiatives regarding the optimization of their layout, since more "user-friendly" platforms promote the integration of older people into services promoted by fintechs in general.

The democratization process, led by development agencies, industries, regulators and private foundations, encourages fintechs to provide a faster path to access digital resources in the financial market that have not yet been explored [115]. These resources are presented similarly to credit facilitation, as loans secured by assets and funding for low-income users. In this sense, fintechs allow access to financial services to be accessible to different social strata.

Additionally, fintechs have proven to be important actors in regard to financial inclusion. Along these lines, Nguyen emphasizes their importance in an emerging economy [116]. He reinforces that the use of financial services through fintechs is linked to perceived financial knowledge, a factor that justifies the usability of fintechs in the contemporary economy. Vasenska et al. highlight this usability during COVID-19, when the applicability of fintechs was crucial to maintaining the health of the financial system [117].

In this way, it is possible to understand that the phase of the dematerialization of access is essential for the institutions in the financial system to develop institutional learning. Consequently, financial market institutions start to develop the necessary know-how to integrate the knowledge of this new technology into the sector. In this stage of institutional learning, institutions place themselves as users, so that they can understand the relevance of this new technological resource. Through this learning, the operational architecture of the financial system begins to evolve gradually.

### 5.2.2. Operational Architecture

In the process of financial system evolution, the architecture of financial operations tends to be modified. In this way, currencies, bonds, properties, assets, commodities, and tokens have begun to be used as exchange methods, even though they are not legally recognized as financial products [118]. In this sense, peer-to-peer (P2P) intermediation platforms have become widely used, becoming a major contribution of fintechs to this sector [84,119]. Therefore, this new model of financial operations represents the introduction of a new set of rules in the market.

The facilities provided by fintechs promote different types of incentives, so that entrepreneurs, investors, and development agencies become part of this market. This feature is important for both developing countries and local economies. Huang highlights the impact of fintechs on small and medium-sized companies [120]. The study demonstrates the importance of the robustness that fintechs represent, which reduces information asymmetry between institutions, improves investment efficiency, and reduces costs.

Through these digitalization incentives, fintechs become a vehicle for the direct intermediation of financial products and services. The elimination of these intermediaries encourages the development of technologies that guarantee the integrity of financial transactions. Cryptoassets, digital currencies, blockchain, digital contracts, and artificial intelligence prove to be faster and cheaper and to have less obstacles in the decentralization of financial operations, promoting a new layout for the financial system [112,121]. Consequently, the consumers start to demonstrate the wide adoption of fintech services with these technological resources, which guarantees the integrity of financial transactions [8].

This factor is responsible for implementing a new system of informal rules in the financial system. Therefore, regulatory institutions are forced to take measures that change the legal status of these operations provided by fintechs. In this study, this movement was called transactional regulation.

### 5.2.3. Transactional Regulation

The modification of informal rules provided by fintechs, after some time, interacts with institutions in the financial system. These institutions become a support instrument for the formalization, adoption, and expansion of new operations provided by fintechs to the financial system. Institutions formalize these rules through a stage called transactional regulation. In this stage, institutions develop a regulatory model based on fintechs to manage their operations. This happens through integration between technologies, platforms, and regulatory authorities.

First, the integration of regulatory technologies in the financial sector has proven to be a path to sustainable development of the financial system. Muganyi et al. state that these technologies support the financial sector, providing access and depth to operations [6]. Additionally, they highlight the role of regtechs as important tools in balancing fintech growth through regulatory imperatives. Regtechs have demonstrated their importance, because there are some fintechs that often operate without formal regulation. Therefore, regtechs ensure operational integrity until there is a social movement for regulatory change [105].

In addition, the fintech phenomenon is responsible for integration between regulatory platforms, which links authorities, suppliers, and consumers. Murinde et al. outlined a future perspective for the financial sector [15]. They suggest that financial system players are focused on developing their own platforms and that through global infrastructures and application programming interfaces, the future of this sector will be shaped. In contrast, Cao et al. assessed whether these investments in digital infrastructure have improved the efficiency of financial operations [18]. They pointed out that the institutions that develop these platforms must strengthen their ability to innovate in order to obtain better performance from their resources.

Finally, the integration of regulatory authorities with financial technologies in their different governance strata has been highlighted. Bin-Nashwan points out that for regulatory authorities to keep up with the digital transformations that took place during

COVID-19, they must incorporate digitized services into their processes, which translates into revenue maximization [122]. Through a contract management measure, regulatory authorities in the financial sector can expand the range of operations of fintechs and allow them to stop operating only as providers of outsourced payment services [123]. In this way, the new financial market rules that emerge from fintechs' activities can become formalized and regulated.

5.2.4. Transactional Efficiency

Subsequently, the evolutionary process of institutional change brought about by fintechs reaches a refining stage. We call this stage transactional efficiency. In this stage, support instruments are developed to ensure the safety of fintech users, reduce transaction costs, and minimize risk. This risk minimization process concerns cost reductions and user accessibility. This accessibility improves customer relationships and the execution of contracts, factors that help to increase the transactional capacity of users and firms in the sector [124,125]. In this sense, the increase in transactional capacity, when encouraged by the institutions within the sector, brings about an increase in the performance of the economy.

Among the growth of economic performance, the sharing economy has demonstrated its validity in explaining future trends. This concept has been approached as a pillar for the determination of a new economic bases in Smart Cities. In this sense, fintechs improve the efficiency of financial operations through risk sharing [126]. Furthermore, they complement the role of intermediation that was traditionally exercised only by banks [119].

Additionally, seeking to improve the efficiency of financial operations, there are measures for the technological management of risks and safety in operations. With advancements in technology, the financial sector has changed its business processes and models, along with its risk control measures [48,112]. With the growth of operational risk brought about by fintechs, institutions in the sector began to promote technological and managerial risk management measures [127]. Then, cloud computing measures, cryptography, and other regtechs entered the market with the aim of ensuring the security of transactions. In addition, in relation to data security, changing business models have forced institutions to adopt measures regarding transaction security [21]. Akartuna et al. highlight the money laundering risks made possible by advances in accounting and payment methods [12]. These threats include cryptocurrencies, transaction laundering, and exclusively digital services. However, with the implementation of countermeasures related to these technologies, the reduced transaction costs and accessibility provided by fintechs promote an increase in economic performance.

The increase in economic performance, as much as it is a consequence of the evolutionary process of institutional change, allows institutions to have more subsidies, promoting an innovation incentive structure. This occurs through the use of regulatory sandboxes that function as a way to promote innovation.

Therefore, throughout this systematic literature review, the dimensions that govern the evolutionary process of institutional change incorporated by fintechs may present cyclical behavior. Consequently, the greater the transactional efficiency, the greater the incentives to dematerialize access. These agents are responsible for carrying out each step in the *evolutionary process of institutional change brought about by fintechs*.

In short, the different evolutionary dimensions presented in this study can be demonstrated to coexist with the institutional analysis undertaken by North [57,59,128]. The process of institutional change brought about by fintechs towards an increase in the performance of the financial system and its dimensions has presented important imperatives for the institutions involved. Similarly, these changes need adaptations by institutions to maintain the health of the financial system. Therefore, we offer three propositions to describe the impact of these dimensions on the financial system and describe how their variables can be detected.

*5.3. Propositions*

In the present literature review, we were able to construct three propositions that describe the evolution of the transactional system associated with fintechs. Within the complexity of this analysis, a theoretical framework was developed that clarifies fintechs as driving agents of a new institutional model of the economy.

Firstly, it is possible to note that the accelerated growth of fintechs has placed institutions in a responsive position towards these innovations. In this sense, the analysis indicates that institutions in the contemporary world are not behaving as protagonists regarding the fintech phenomenon. Therefore, institutional operations integrate fintech resources in response to external pressure and not in a preventive way. Fintechs have been responsible for a new way of carrying out transactions, due to the compatibility of electronic devices and high-capacity servers [129]. As a consequence, consumers have sought intermediation through digital resources, and institutions have been required to operate in this sense.

The institutional model proposed by fintechs is based on an automated economy, which drives institutional changes [21]. However, this evolutionary change represents a challenge associated with linking these technological resources to users. In this way, fintechs have been responsible for a decentralized financial model in which the regulatory framework is supported by these innovations. Additionally, it is up to fintechs to offer solutions regarding uncertainties in transactions [13]. Currently, institutions still offer stopgap measures to deal with problems stemming from the emergence of fintechs, since many problems have appeared as the sector develops [80]. As a consequence of the challenges imposed by fintech development and the institutions' reactive stance, we present the following proposition:

**Proposition 1.** *The speed of technological evolution in the financial system is higher than that of institutional evolution.*

Furthermore, fintechs are responsible for changing the level of financial services, leading to a more dynamic and responsive institutional model. This new model of the institutional matrix stems from a reduction of state powers. Thus, fintechs force the institutional matrix to operate informally, based on market principles [1]. In this new model, technologies such as blockchain and big data, artificial intelligence, and biometry establish themselves as tools that support the modification of operations within the financial system [16].

The services provided by fintechs to their clients bring about concepts linked to the mobile payments model. Furthermore, self-service resources and branchless operations bring comfort to clients and competitivity to business models [21]. IT resources geared towards clients demonstrate that the service model proposed by fintechs gives users more security [81]. As such, the services provided by fintechs have increased on the side of consumers, since researchers have noted that trust, financial literacy, and transparency are factors that directly influence the adoption of fintechs' services [65,70].

In this sense, the organizations that do not adapt to the financial system's new modus operandi will become obsolete. In this way, it is necessary for the institutions of the financial system to incorporate big data, machine learning, and blockchain resources for information storage. These resources have been used as tools for improving service delivery and consumer data protection for decentralized transactions. As these financial providers grow, fintechs face institutional challenges correlated with security, money laundering, and privacy [1]. Thus, we have made the following proposition:

**Proposition 2.** *Fintechs cause changes in the financial system's institutional matrix.*

Finally, it was possible to observe that institutional matrices with higher levels of uncertainty offer greater opportunities for fintechs. Thus, post-crisis regulatory scenarios with systemic risks and obsolete regulatory concepts have been shown to be the ideal envi-

ronment for fintech implementation [11,98]. However, political and economic organizations need to adapt their modes of operation to this new model brought about by fintechs.

During the financial crisis, political and economic institutions needed to use technological and regulatory resources to respond to many corporate scandals [130]. In this sense, Yang and Li point out that fintechs emerged, providing technology-based regulations [98]. These regulatory resources focus on data monitoring and can provide solutions to the inefficiency and ineffectiveness of traditional financial regulations.

Anagnostopoulos points out that the regulatory standards brought about by fintechs serve as a base for political and economic organizations [80]. This occurs due to the fact that fintechs mitigate uncertainty related to transaction transparency and facilitate supervision by the financial market's agents. Therefore, with the growth of activities provided by fintechs, the need for national governance actors in the market decreases [131]. This factor defines a new institutional model that modifies the standards for the regulation of financial transactions in a more holistic way [132]. Thus, we can propose that:

**Proposition 3.** *Fintechs cause a change in institutional rules in response to uncertainties, and political and economic organizations must adapt to this situation.*

In short, the technology generated by fintechs has aroused the trust of users and these decentralized platforms can potentially provide a new basis for decentralized business models [13]. Haddad and Hornuf investigated the economic and technological determinants for the incorporation of fintechs in financial systems and found that countries with greater available capital tended to witness the increased formation of technology startups [64]. However, the rigid regulatory barriers imposed by countries with a solid institutional structure can jeopardize the commercial sustainability of these platforms, with a lack of adequacy on the part of the authorities involved [24].

*5.4. Research Agenda*

In this systematic review, we identified the directions for future research regarding fintechs and institutions. Within each dimension that was analyzed, it was possible to observe that the literature lacks a deep analysis, both of the dimensions and of their variables. In this sense, we subdivided these new themes that relate to the four dimensions into perspectives involving 'business models', 'effects on the financial ecosystem', 'financial operation engineering', and 'regulatory structures'.

Regarding the 'business models' perspective, researchers highlighted that fintechs support the appearance of new business models which disruptively modify financial services. These models include payment services, asset management, collective financing, loans, insurance, and the capital market. Furthermore, in relation to this perspective, there are more specific sub-themes, such as customer relationships, investment management, regulation, and security. Within these sub-themes, there is the evaluation of the appearance and strategic operations of startups in this sector. In this way, some questions emerge, such as: Which company and market variables moderate the relationship between fintechs and performance? How can the digital resources provided by fintechs facilitate the transformation of financial operations? Which metrics are important to measure the growth of digital platforms?

As for the 'effects of fintechs on the financial ecosystem', the literature has focused on the process of providing technology to the consumers of financial products. Considering this process, it is important to note the impact it has on the intermediaries of financial operations and on the sector's authorities. Additionally, other themes geared towards the impact on users stand out, including subjects such as digital inclusion, financial literacy, and incentives from authorities in the sector. Thus, some questions arise: What is the level of digital inclusion of fintech users? How can one encourage the use of digital resources provided by fintechs? Furthermore, there is support for analyses regarding the impact of fintechs on the stability of financial systems and their regulatory inflection points.

These subjects are associated with the consolidation of fintechs in countries with weaker or more robust regulatory structures, and highlight the adaptations undergone by the regulation processes.

When the perspective is geared toward 'financial operation engineering', it is possible to note that the organizations that oppose fintechs in the market have reinforced their attention in regard to their operations. Subjects that focus on financial transfers between countries with different currencies are trending, with the intent of reducing expenses with exchange taxes. Thus, there are pertinent questions regarding how firms should use fintech resources to increase their efficiency, reduce costs, and conduct transactions with other countries. Other questions are associated with the security provided by blockchain platforms and the amount of resources that can be incorporated, interlinking the fintech ecosystem with the internet of things (IoT). Finally, strategies are being evaluated to institutionalize cryptocurrency in order to preserve institutional control in the economies of different countries.

Finally, the perspective of 'fintechs and regulatory structures' demonstrated an important role for the studies that integrate the viewpoints of fintechs and institutions. This perspective emphasizes details related to the supervision and regulation of financial operations. These details relate to measuring the financial operations conducted over digital platforms and the regulation that controls these operations. This perspective includes the development of regtechs, which help authorities to maintain the financial system's health. Furthermore, studies have evaluated how fintechs can provide intermediate operations between countries with different regulatory structures and their respective impacts. This topic includes a series of questions that remain to be answered, such as: How can sector authorities develop resources to regulate digitalized financial operations? What are the strategies used by regulatory institutions to ensure the legality of financial operations? What is the frontier of action of regulatory authorities in the face of innovations provided by fintechs? These subjects will generate positive conclusions for the scientific community pertaining to 'fintechs and institutions'. We recommend that a higher number of studies should be conducted in order to validate this review's determinants empirically.

## 6. Final Considerations

In the present study, we aimed to review and analyze the current state of the research that correlates fintechs and institutions. To fulfill this objective, a systematic literature review was conducted on fintechs and institutions. This study provides three contributions. The first is the creation of a framework that describes the change in the institutional model proposed by fintechs. This framework is based on an adaptive perspective on the financial and regulatory institutions regarding fintechs. This framework clarifies how fintechs modify institutional behavior regarding service provision.

The second contribution is the formulation of three theoretical propositions that justify institutional behavior in light of the evolutionary process of fintechs. This process is detailed, identifying the need for a reformulation of the financial system's institutional model due to the incorporation of these technologies. This work is unprecedented and provides a broad view of how the scientific literature has described this evolution.

Finally, this study's third contribution is the identification of new research directions pertaining to fintechs and institutions. With the new technological concepts introduced by fintechs, it is expected that the evolution of this market will reach significant levels in the next few years. In this sense, the topics presented here allow the reader to trace the future perspectives regarding the behavior of the institutional economy.

Regarding the implications of this study on public policy, the importance of this topic for the implementation of measures aimed at digital governance is noted. These measures can guarantee savings for countries in terms of diluting operational risks, facilitating financial investments and allocating resources. In this way, it is already possible to measure how the regulatory architecture of these countries can be managed for better implementation.

Regarding the managerial implications, the studies has been focused on the administrative and financial sectors. Within the area of administrative interests, this review can help managers with the elaboration of competitiveness initiatives, the implementation of technology in a hierarchical structure, the adaptation to technological regulation, and in the development of an information apparatus aimed at IT governance within organizations. Additionally, the studies focused on financial field refer to opportunities for shared credit, crowdfunding, international transfers, and new opportunities to reduce transaction costs.

Further research is recommended, which can validate the determinants evaluated in this review empirically and which can elucidate this new institutional structure. We suggest evaluating the proposed framework in different institutional contexts, ranging from developed economies to emerging economies. Exploratory and descriptive research based on this framework can be part of a future research agenda.

As for this study's limitations, the documents that were reviewed depend on the databases that were used. Consequently, important documents may have been left out of this study. Another limitation of this study is the time period that was chosen, since it could be adjusted at the researcher's discretion.

**Author Contributions:** Conceptualization, J.T.-G. and D.C.-T.; methodology, J.T.-G. and D.C.-T.; data curation, D.C.-T.; writing—original draft preparation, J.T.-G., D.C.-T., A.A.L., M.H.-M. and J.R.; writing—review and editing, supervision, J.T.-G., D.C.-T., A.A.L., M.H.-M. and J.R. All authors have read and agreed to the published version of the manuscript.

**Funding:** CAPES—Coordenação de Aperfeiçoamento de Pessoal de Nível Superior.

**Institutional Review Board Statement:** The study was conducted in accordance with the Declaration of Helsinki, and approved by the Institutional Review Board.

**Informed Consent Statement:** Not applicable.

**Conflicts of Interest:** The authors declare no conflict of interest.

## Appendix A

**Table A1.** Documents found in the systematic literature review.

| N | Title | Authors | Year |
|---|---|---|---|
| 1 | *A 2020 perspective on "A fair contract signing protocol with blockchain support"* | Josep-Lluis Ferrer-Gomila, Maria Francisca Hinarejos | 2020 |
| 2 | *A Tale of Two Markets: How Lower-end Borrowers Are Punished for Bank Regulatory Failures in Nigeria* | Philemon Omede | 2020 |
| 3 | *A truly future-oriented legal framework for fintech in the EU.* | Kapsis, I. | 2020 |
| 4 | *Archiving and digitizing of customer records of golden rural bank of the Philippines, Inc.* | Princess May Subia, Reynaldo Corpuz | 2020 |
| 5 | *Artificial intelligence and automation in financial services: The case of Russian banking sector* | Goncharenko, Andrea Miglionico | 2020 |
| 6 | *Bank financial capability on MSME lending amid economic change and the growth of Fintech companies in Indonesia* | Martino Wibowo, Vesarach Aumeboonsuke | 2020 |
| 7 | *Banking and regulatory responses to fintech revisited—Building the sustainable financial service 'ecosystems' of tomorrow.* | Fenwick, M., & Vermeulen, E. P. | 2020 |
| 8 | *Banking goes digital: The adoption of FinTech services by German households* | Moritz Jünger, Mark Mietzner | 2020 |
| 9 | *Banking on Blockchain: An Evaluation of Innovation Decision Making* | Priya Dozier, Troy Montgomery | 2020 |
| 10 | *Banking sector earnings management using loan loss provisions in the Fintech era* | Peterson Ozili | 2020 |
| 11 | *Blockchain and insurance: a review for operations and regulation* | Richard Brophy | 2020 |

**Table A1.** *Cont.*

| N | Title | Authors | Year |
|---|-------|---------|------|
| 12 | *Blockchain disruption and decentralized finance: The rise of decentralized business models* | Yan Chen, Cristiano Bellavitis | 2020 |
| 13 | *Concealed Risks of FinTech and Goal-Oriented Responsive Regulation: China's Background and Global Perspective.* | Donggen, X. U., & Dawei, X. U. | 2020 |
| 14 | *Conventional banks and Fintechs: how digitization has transformed both models* | Elisabeth Paulet, Hareesh Mavoori | 2020 |
| 15 | *Cooperative financial institutions: A review of the literature* | Donal McKillop, Declan French, Barry Quinn, Anna Sobiech, John Wilson | 2020 |
| 16 | *Decentralized finance* | Dirk Zetzsche, Douglas Arner, Ross Buckley | 2020 |
| 17 | *Digital cubic space as a new economic augmented reality* | Natalia Kraus, Kateryna Kraus, Andrusiak | 2020 |
| 18 | *Emergent role of fintech in financial landscape: A perspective on banking industry* | Kumar, Agrawal, Aliza | 2020 |
| 19 | *Financial inclusion research around the world: A review* | Peterson Ozili | 2020 |
| 20 | *Fintech and Financial Stability Potential Influence of FinTech on Financial Stability, Risks and Benefits* | Milena Vučinić | 2020 |
| 21 | *Fintech in financial reporting and audit for fraud prevention and safeguarding equity investments* | Paulina Roszkowska | 2020 |
| 22 | *FinTech, blockchain and Islamic finance: An extensive literature review* | Mustafa Rabbani, Shahnawaz Khan, Eleftherios Thalassinos | 2020 |
| 23 | *Fintech, financial inclusion and income inequality: a quantile regression approach* | Ayse Demir, Vanesa Pesqué-Cela, Yener Altunbas, Victor Murinde | 2020 |
| 24 | *Fintech: research directions to explore the digital transformation of financial service systems* | Christoph Breidbach, Bryon Keating, Chiehyeon Lim | 2020 |
| 25 | *From shadow banking to digital financial inclusion: China's rise and the politics of epistemic contestation within the financial stability board* | Peter Knaack, Julian Gruin | 2020 |
| 26 | *Governing the gold rush into emerging markets: a case study of Indonesia's regulatory responses to the expansion of Chinese-backed online P2P lending* | Angela Tritto, Yujia He, Victoria Junaedi | 2020 |
| 27 | *Granting access to real-time gross settlement systems in the fintech era* | Bagio Bossone, Gynedi Srinivas, Holti Banka | 2020 |
| 28 | *How individual investors react to negative events in the fintech era? Evidence from China's peer-to-peer lending market* | Xueru Chen, Xiaoji Hu, Shenglin Ben | 2020 |
| 29 | *Impact of customers' digital banking adoption on hidden defection: A combined analytical–empirical approach* | Yoonseock Son, Hyeokko Kwon, Giri Tayi, Wonseok Oh | 2020 |
| 30 | *Industry 4.0 in finance: the impact of artificial intelligence (ai) on digital financial inclusion* | David Mhlanga | 2020 |
| 31 | *Initial coin offerings (ICOs): Benefits, risks and success measures* | Alfreda Šapkauskienė, Ingrida Višinskaitė | 2020 |
| 32 | *Legal Governance on Fintech Risks: Effects and Lessons from Chinae* | Yuan, K., & Duoqi, X. U | 2020 |
| 33 | *Mobile money adoption and usage and financial inclusion: mediating effect of digital consumer protection* | Okello Candiya Bongomin, G.,Ntayi | 2020 |
| 34 | *New quality of financial institutions and business management* | Nataliia Kraus, Kateryna Kraus, Valerii Osetskyi | 2020 |
| 35 | *Not Just Another Shadow Bank: Chinese Authoritarian Capitalism and the 'Developmental' Promise of Digital Financial Innovation* | Julian Gruin, Peter Knaack | 2020 |

**Table A1.** *Cont.*

| N | Title | Authors | Year |
|---|---|---|---|
| 36 | *Regulating Fintech in the EU: the Case for a Guided Sandbox.* | Ringe, W. G., & Christopher, R. U. O. F. | 2020 |
| 37 | *Regulation and Recent Trends in High-Interest Credit Markets.* | Malone, C., & Skiba, P. M. | 2020 |
| 38 | *Regulatory Technology: Replacing Law with Computer Code* | Eva Micheler, Anna Whaley | 2020 |
| 39 | *Responsible AI-based Credit Scoring–A Legal Framework.* | Langenbucher, K | 2020 |
| 40 | *Risk spillovers between FinTech and traditional financial institutions: Evidence from the U.S.* | Jianping Li, Jingyu Li, Xiaoqian Zhu, Yinhong Yao, Barbara Casu | 2020 |
| 41 | *Technology v Technocracy: Fintech as a Regulatory Challenge* | Saule Omarova | 2020 |
| 42 | *The data sharing paradox: BigTechs in finance* | Oscar Borgogno, Giuseppe Colangelo | 2020 |
| 43 | *The Development and Regulation of Cryptoassets: Hong Kong Experiences and a Comparative* | Robin Huang, Demin Yang, Ferdinand Loo | 2020 |
| 44 | *The Disruptive Effect of Distributed Ledger Technology and Blockchain in the over the counter derivatives market.* | Paolini, A. | 2020 |
| 45 | *The effects of eliminating Riba in foreign currency transactions by introducing global FinTech network* | Mohammad Selim | 2020 |
| 46 | *The European Union Proposal for a Regulation on Cross-Border Crowdfunding Services: A Solemn or Pie-Crust Promise?* | Staikouras, P | 2020 |
| 47 | *The impact of the revised payment services directive on the market for payment initiation services* | Bruno Yawe, Ibrahim Mukisa | 2020 |
| 48 | *The Innovation Research of Contract Farming Financing Mode under the Block Chain Technology* | Dehua Zhang | 2020 |
| 49 | *The payment systems revolution: India's story* | Narendra Kumar, Abhishek Thakur, Raghuraj, Lalit Mohan | 2020 |
| 50 | *The Promise and Perils of Insurtech.* | Lin, L., & Chen, C. C. | 2020 |
| 51 | *The regulation of crypto-assets in the EU–investment and payment tokens under the radar.* | Ferrari, V. | 2020 |
| 52 | *The rise and rise of financial technology: The good, the bad, and the verdict* | Nofie Iman, N. | 2020 |
| 53 | *The Risks of Mobile Payment and Regulatory Responses: A Hong Kong Perspective.* | Huang, R. H., Cheung, C. S. W., & Wang, C. M. L | 2020 |
| 54 | *The road to RegTech: the (astonishing) example of the European Union* | Ross Buckley, Douglas Arner, Dirk Zetzsche, Rolf Weber | 2020 |
| 55 | *The small-dollar loan industry: a new era of regulatory reform—and emerging competition?* | Thomas Hemphill | 2020 |
| 56 | *Transformation needed—report on the 6th international conference on credit risk analysis and management* | Simone Westerfeld, Beatrix Wullschleger | 2020 |
| 57 | *Twenty-first Century Financial Regulation: P2P Lending, Fintech, and the Argument for a Special Purpose Fintech Charter Approach.* | Luther, J. | 2020 |
| 58 | *What have we learnt from 10 years of fintech research? a scientometric analysis* | Jiajia Liu, Xuerong Li, Shouyang Wang | 2020 |
| 59 | *What's in the "Black Box"? Balancing Financial Inclusion and Privacy in Digital Consumer Lending.* | Chou, A. | 2020 |
| 60 | *A fair contract signing protocol with blockchain support* | Josep-Lluis Ferrer-Gomila, Francisca Hinarejos, Andreu-Pere Isern-Deyà | 2019 |
| 61 | *A great leap of faith: the cashless agenda in Digital India.* | Athique, A. | 2019 |

**Table A1.** *Cont.*

| N | Title | Authors | Year |
|---|---|---|---|
| 62 | *Competition and stability in modern banking: A post-crisis perspective* | Xavier Vives | 2019 |
| 63 | *Credit intermediation and the European internal market for mortgage credit* | Diederik Bruloot, Evariest Callens, Michiel De Muynck | 2019 |
| 64 | *Cross-border regulation and fintech: are transnational cooperation agreements the right way to go?* | Ivanova, P. | 2019 |
| 65 | *Dematerialization of banking products and services in the digital era* | Shahrazad Hadad, Constantin Bratianu | 2019 |
| 66 | *Digital Payments: Impact Factors and Mass Adoption in Sub-Saharan Africa* | Leigh Soutter, Kenzie Ferguson, Michael Neubert | 2019 |
| 67 | *Do digital technologies have the power to disrupt commercial banking?* | Golubić, G | 2019 |
| 68 | *Encouraging Entrepreneurship and Economic Growth* | David Ahlstrom, Amber Chang, Jessie Cheung | 2019 |
| 69 | *FinTech on the dark web: The rise of cryptos.* | Todorof, M. | 2019 |
| 70 | *FinTech sector and banking business: competition or symbiosis?* | Mikhail Zveryakov, Sergii Sheludko, Elena Sharah, Victoria Kovalenko | 2019 |
| 71 | *Fintechs: A literature review and research agenda* | Eduardo Milian, Mauro de Spinola, Marly de Carvalho | 2019 |
| 72 | *Following the cyber money trail: Global challenges when investigating ransomware attacks and how regulation can help* | Angela Irwin, Caitlin Dawson, | 2019 |
| 73 | *Funds sharing regulation in the context of the sharing economy: Understanding the logic of China's P2P lending regulation* | Tao Yu, Wei Shen | 2019 |
| 74 | *Global Financial Regulation: Shortcomings and Reform Options* | Emily Jones, Peter Knaack | 2019 |
| 75 | *Mind the gap: the consideration of financial technologies and blockchain in the reform of the Vertical Agreements Block Exemption Regulation.* | Chambers, L. M. | 2019 |
| 76 | *Public Financial Law and digital economy.* | Tsindeliani, I. | 2019 |
| 77 | *Regulatory Fitness: Fintech, Funny Money, and Smart Contracts* | Roger Brownsword | 2019 |
| 78 | *Regulatory Sandboxes.* | Allen, H. J. | 2019 |
| 79 | *Success factors in Title III equity crowdfunding in the United States* | Stanislav Mamonov, Ross Malaga | 2019 |
| 80 | *The influence of financial innovations on eu countries banking systems development* | Oleksiy Druhov, Vira Druhova, Olena Pakhnenko | 2019 |
| 81 | *Virtual and cryptocurrencies—regulatory and anti-money laundering approaches in the European Union and in Switzerland* | Frick, T. A. | 2019 |
| 82 | *Complacency, capabilities, and institutional pressure: understanding financial institutions' participation in the nascent mobile payments ecosystem* | Kui Du | 2018 |
| 83 | *Cooperative banking and digital transformation: towards a new relationship model with members and clients* | Ricardo Palomo Zurdo, Yakira Fernández Torres, Milagros Gutiérrez Fernández | 2018 |
| 84 | *Cross-Border Crowdfunding: Towards a Single Crowdlending and Crowdinvesting Market for Europe* | Dirk Zetzsche, Christina Preiner | 2018 |
| 85 | *Determinants of the financial services market functioning in the era of the informational economy development* | Serhiy Shkarlet, Maksym Dubyna, Olena Zhuk | 2018 |
| 86 | *Dialectic tensions in the financial markets: a longitudinal study of pre- and postcrisis regulatory technology* | Wendy L. Currie, Daniel Gozman, Jonathan Seddon | 2018 |
| 87 | *Evolutionary Approaches and the Construction of Technology-Driven Regulations* | Dong Yang & Min Li | 2018 |
| 88 | *Financial-return Crowdfunding and Regulatory Approaches in the Shadow Banking, FinTech and Collaborative Finance Era* | Eugenia Macchiavello | 2018 |

**Table A1.** *Cont.*

| N | Title | Authors | Year |
|---|---|---|---|
| 89 | *Fintech and regtech: Impact on regulators and banks* | Ioannis Anagnostopoulos | 2018 |
| 90 | *Fintech and the Future of the Payment Landscape: The Mobile Wallet Ecosystem A Challenge for Retail Banks?* | Anna Eugenia Omarini | 2018 |
| 91 | *Fintech ecosystem and landscape in Russia* | Vladimir Soloviev | 2018 |
| 92 | *Fintech risk management: A research challenge for artificial intelligence in finance* | Paolo Giudici | 2018 |
| 93 | *Fintech Venture Capital* | Douglas Cumming, Armin Schwienbacher, | 2018 |
| 94 | *Fintech: Ecosystem, business models, investment decisions, and challenges* | In Lee, Yong Jae Shin | 2018 |
| 95 | *From the Institutional to the Platform Economy* | Aleksandr Sukhodolov, Yury Beryozkin | 2018 |
| 96 | *Impact of digital finance on financial inclusion and stability* | Peterson Ozili | 2018 |
| 97 | *Information, incentives, and effects of risk-sharing on the real economy* | Mark Liu, Wenfeng Wub, Tong Yu | 2018 |
| 98 | *Institutional Changes And Ditigalization Of Business Operations In Financial Institutions* | Elena Tarkhanova, Elena Chizhevskaya, Natalia Baburina | 2018 |
| 99 | *Investor Platform Choice: Herding, Platform Attributes, and Regulations* | Yang Jiang, Yi-Chun (Chad) Ho, Xiangbin Yan, Yong Tan | 2018 |
| 100 | *Modeling of FinTech market development (on the example of Ukraine* | Alina Bukhtiarova, Arsen Hayriyan, Nikol Bort, Andrii Semenog | 2018 |
| 101 | *Propensity of contracting loans services from FinTech's in Brazil* | Luis Hernan Contreras-Pinochet, Guilherme Tongnole Diogo, Evandro Luiz Lopes, Eliane Herrero, Ricardo Luiz Pereira Bueno | 2018 |
| 102 | *The emergence of the global fintech market: economic and technological determinants* | Christian Haddad, Lars Hornuf | 2018 |
| 103 | *The emerging Cloud Dilemma: Balancing innovation with cross-border privacy and outsourcing regulations* | Daniel Gozman, Leslie Willcocksc | 2018 |
| 104 | *The Impact of Selected Regulations on the Development of Payments Systems in Poland* | Mateusz Folwarski | 2018 |
| 105 | *The influence of financial technologies on the global financial system stability* | Galyna Azarenkova, Iryna Shkodina, Borys Samorodov, Maksym Babenko, Iryna Onishchenko | 2018 |
| 106 | *The opportunities of engaging FinTech companies into the system of crossborder money transfers in Ukraine* | Yuriy Petrushenko, Liudmyla Kozarezenko, Aldona Glinska-Newes, Maryna Tokarenko, Maryna But | 2018 |
| 107 | *The Payment Services Directive II and Competitiveness: The Perspective of European Fintech Companies* | Inna Romānova, Simon Grima, Jonathan Spiteri, Marina Kudinska | 2018 |
| 108 | *The Regulation of Initial Coin Offerings in China: Problems, Prognoses and Prospects* | Hui Deng, Robin Hui Huang, Qingran Wu | 2018 |
| 109 | *Catching up with Indonesia's fintech industry* | Kevin Davis, Rodney Maddock, Martin Foo | 2017 |
| 110 | *Digital Finance and FinTech: current research and future research directions* | Peter Gomber, Jascha-Alexander Koch, Michael Siering | 2017 |
| 111 | *Fintech as financial innovation—The possibilities and problems of implementation* | Svetlana Saksonova, Irina Kuzmina-Merlino | 2017 |

**Table A1.** *Cont.*

| N | Title | Authors | Year |
|---|---|---|---|
| 112 | *FinTech, RegTech, and the Reconceptualization of Financial Regulation* | Douglas Arner, Jànos Barberis, Ross Buckley | 2017 |
| 113 | *Fintech, Regulatory arbitrage, and the rise of shadow banks* | Greg Buchak, Gregor Matvos, Tomasz Piskorski, Amit Seru | 2017 |
| 114 | *From digital currencies to digital finance: the case for a smart financial contract standard* | Willi Brammertz, Allan I. Mendelowitz | 2017 |
| 115 | *Future living framework: Is blockchain the next enabling network?* | Maria-Lluïsa, Marsal-Llacuna | 2017 |
| 116 | *Payment innovations in Poland: a new approach of the banking sector to introducing payment solutions* | Michał Polasik, Dariusz Piotrowski | 2017 |
| 117 | *The digital revolution in financial inclusion: international development in the fintech era* | Daniela Gabor, Sally Brooks | 2017 |
| 118 | *The transition from traditional banking to mobile internet finance: an organizational innovation perspective—a comparative study of Citibank and ICBC* | Zhuming Chen, Yushan Li, Yawen Wu, Junjun Luo | 2017 |
| 119 | *Entry of FinTech Firms and Competition in the Retail Payments Market* | Jooyong Jun, Eunjung Yeo | 2016 |
| 120 | *FinTech in China: From Shadow Banking to P2P Lending* | Jànos Barberis, Douglas Arner | 2016 |
| 121 | *FinTech in Taiwan: a case study of a Bank's strategic planning for an investment in a FinTech company* | Jui-Long Hung, Binjie Luo | 2016 |
| 122 | *New factors inducing changes in the retail banking customer relationship management (CRM) and their exploration by the FinTech industry* | Marcin Kotarba | 2016 |
| 123 | *Payment innovations in poland: the role of payment services in the strategies of commercial banks* | Michal Polasik, Dariusz Piotrowski | 2016 |

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
