# Peer review of "Fintechs and Institutions: A Systematic Literature Review and Future Research Agenda"

_jtaer, doi:10.3390/jtaer17020038_

Round 1

Reviewer 1 Report

Dear authors,

I have several recommendations for you:

  1. Fully depersonalize your manuscript.
  2. I suggest these references to enrich your study:

    doi: 10.3390/risks9030048

    doi: 10.29036/jots.v12i23.275

    doi:10.22381/RCP19202010

  3. Typical Discussion is missing. It is a crucial part of the manuscript. Compare your results to similar studies.
  4. Detete the label of chapter 6.1. Study limitations, let only the paragraph in Conclusion.

Good luck in your future work.

Author Response

Dear reviewer 1.

We are grateful for your comments to improve our work. As requested, we apply all recommendations to our article.

Initially, as per comment 1, we depersonalized the article before making the changes. In the event that the co-authorship of the article is appearing, it may be a subject of the submission system, because we do not put the co-authorship on the article.

We appreciate the suggested references. The same have been incorporated throughout the text.

Finally, we deleted the label of chapter 6.1.

Reviewer 2 Report

The paper concerns an important topic, and is well presented. There are two main weaknesses.

The first concerns the search methodology: the authors exclude from the search papers that are not in the categories "business, management and accounting" or "Economics, Econometrics and Finance". Doing so, however, they miss out important contributions; for example from Decision sciences: operational research,  Information systems and statistics. This explains why, for example, much of the literature on Fintech Risk Management (for a review see e.g. Giudici P. (2018) Fintech Risk Management: A Research Challenge for Artificial Intelligence in Finance. Front Artif Intell. 2018 Nov 27;1:1. ). The authors should include this literature.

The second weakness concerns the four dimensions in which results are categorised, summarised in Table 3. It would help to explain the rationale of this categorisation a bit better. For example, the first and second dimension are "macro" dimensions: one about the external environment, one about the fintech internal environment; whereas the third and fourth dimensions are "macro" dimensions, referred to the transactional level, either from an external or an internal view point. By improving the categorisation, more papers should be included that concern the external environment, either from a macro or a micro perspective. For example, the issue of cyber risks,  operational risks and market integrity/fraud detection, which particularly involve the micro perspective, should be included. See for example the paper by Aldasoro et al, (2022) the drivers of cyber risk., Bank for International settlements working paper and journal of financial stability.

The Auhors should try to answer to he above queries.

Author Response

Dear reviewer 2

We are grateful for the comments. They were very helpful in improving the capability of our document. Based on Comment1, we evaluated the literature that interconnects fintech topics and institutions and, according to your recommendation, we decided to add the research areas: “decision sciences” and “social sciences”. These dimensions led us to increase the number of documents analyzed and to have a broader perspective on the topic. Due to the increase in new literature (“decision sciences” and “social sciences”), we improved the results of the document and inserted a discussion sector in the analysis of the result that explained in more detail the relevance of the work. However, we did not add topics related to “Computer science” or “Engineering”, due to the belief that this was not the purpose of our analysis. As we sought to understand the perspectives of institutions, we were dealing with a topic with implications more focused on public policies, accounting, management and social issues. We highlight this last in the method.

Regarding this commentary, we have added these implications in the conclusion to clarify the focus of the work. Soon, we added the recommended themes. As for comment 2, in the discussion we made an assessment of the macro and micro. However, the dimensions of this issue cover the entire financial and institutional system. Therefore, we were able to understand the action of users and support institutions, regarding their posture of governing and collaborating with the emergence of fintechs. However, we decided to implement the discussion in a simpler way so that the objectives of the theme are presented more clearly.

Best regards

Reviewer 3 Report

Fintechs and institutions: A systematic literature review and future research agenda

This is a review article in which the authors make a comprehensive literature review related to fintechs in order to shed light on the aspects debated and clarified so far in the literature related to this topic and also to point where research is heading. One focus is on the links between fintech and institutions.

The paper adds to the literature in three ways: the description of a framework that describes the change in the institutional model proposed by the fintechs; the formulation of three theoretical propositions that justify institutional behavior in light of the fintechs evolutionary process; the identification of new research directions pertaining to fintechs and institutions.

The paper is coherent, well written and provides well-constructed arguments.

The method used is well described and adequate. The recommendations and conclusions are interesting, sound and provide a good addition to the existing literature.

The manuscript is at an advanced stage.

Overall, an interesting work.

Author Response

Dear reviewer 3

We are very happy with your comments about our work. We work hard to deliver our best. In this sense, we seek to improve all the sectors described to improve the article as a whole.

Best regards

Round 2

Reviewer 2 Report

The authors have included some of my suggestions.
It remains to include a discussion on cyber risks and operational risks which typically affect

Please insert comments on papers that review these issues , such as:

Aldasoro et al, (2022) “the drivers of cyber

risk., Journal of financial stability.

and

Giudici and Bilotta (2004), modelling

operational losses: a Bayesian approach , quality and 

reliability engineering international